# Metabolism and Health Effects of Rare Sugars in a CACO-2/HepG2 Coculture Model

**DOI:** 10.3390/nu14030611

**Published:** 2022-01-30

**Authors:** Amar van Laar, Charlotte Grootaert, Filip Van Nieuwerburgh, Dieter Deforce, Tom Desmet, Koen Beerens, John Van Camp

**Affiliations:** 1Department of Food Technology, Safety & Health, Faculty of Bioscience Engineering, Ghent University, Coupure Links 653, 9000 Ghent, Belgium; amar.vanlaar@ugent.be (A.v.L.); charlotte.grootaert@ugent.be (C.G.); 2NXTGNT, Faculty of Pharmaceutical Sciences, Ghent University, Ottergemsesteenweg 460, 9000 Ghent, Belgium; Filip.VanNieuwerburgh@UGent.be (F.V.N.); dieter.deforce@ugent.be (D.D.); 3Centre for Synthetic Biology, Department of Biotechnology, Faculty of Bioscience Engineering, Ghent University, Coupure Links 653, 9000 Ghent, Belgium; tom.desmet@ugent.be (T.D.); koen.beerens@ugent.be (K.B.)

**Keywords:** cell, digestion, Caco-2/HepG2 model, gene expression, liver fat, rare sugars

## Abstract

Non-alcoholic fatty liver disease (NAFLD) has become the most prevalent liver disease worldwide and is impacted by an unhealthy diet with excessive calories, although the role of sugars in NAFLD etiology remains largely unexplored. Rare sugars are natural sugars with alternative monomers and glycosidic bonds, which have attracted attention as sugar replacers due to developments in enzyme engineering and hence an increased availability. We studied the impact of (rare) sugars on energy production, liver cell physiology and gene expression in human intestinal colorectal adenocarcinoma (Caco-2) cells, hepatoma G2 (HepG2) liver cells and a coculture model with these cells. Fat accumulation was investigated in the presence of an oleic/palmitic acid mixture. Glucose, fructose and galactose, but not mannose, l-arabinose, xylose and ribose enhanced hepatic fat accumulation in a HepG2 monoculture. In the coculture model, there was a non-significant trend (*p* = 0.08) towards higher (20–55% increased) median fat accumulation with maltose, kojibiose and nigerose. In this coculture model, cellular energy production was increased by glucose, maltose, kojibiose and nigerose, but not by trehalose. Furthermore, glucose, fructose and l-arabinose affected gene expression in a sugar-specific way in coculture HepG2 cells. These findings indicate that sugars provide structure-specific effects on cellular energy production, hepatic fat accumulation and gene expression, suggesting a health potential for trehalose and l-arabinose, as well as a differential impact of sugars beyond the distinction of conventional and rare sugars.

## 1. Introduction

The prevalence of obesity is rapidly increasing and is accompanied by an increase in type 2 diabetes and non-alcoholic fatty liver disease (NAFLD), which currently affect about 1 in 11 (diabetes) and 1 in 4 (NAFLD) people on a global scale [1,2,3]. NAFLD is a disease in which at least 5% of the liver tissue is composed of fat. In its early stage of simple steatosis, NAFLD is reversible, although factors such as oxidative stress, inflammation and insulin resistance can cause progression towards hepatic steatohepatitis at a later, more severe, stage of NAFLD [4,5]. In this context, high intake of free sugars is of concern, as epidemiological evidence suggests a relationship between excessive sugar consumption and development of obesity [6]. Furthermore, it has been suggested that fructose, in particular, is a major mediator of hepatic fat accumulation [7]. 

‘Rare sugars’, which are defined by the International Society of Rare Sugars as ‘monosaccharides and their derivatives that are present in limited quantities in nature’, may be promising sugar replacers, based on studies reporting low caloric content and reduced glycemic responses for rare sugars, such as allulose and tagatose, compared to conventional sugars [8,9]. Recently, rare sugars also attracted attention due to innovative biocatalysis strategies that allow efficient production of these otherwise insufficiently available sugars, including biosynthesis of the rare glucobioses kojibiose [10] and nigerose [11] (visualized in Figure 1) from widely available sugars using (engineered) sucrase phosphorylase [12,13]. These innovations allow further research on the metabolic health impact of rare sugars, which is still in an early stage, especially for rare disaccharides [14]. 

Sugars are digested and absorbed primarily at the level of the intestinal brush border, where brush border enzymes, such as sucrase-isomaltase, maltase-glucoamylase, lactase-phlorizin hydrolase and trehalase, act on multiple sugars with varying affinity, resulting in differences in digestion speed and hence metabolic impact [15,16]. The digestion rate is affected further by epigenetic modifications, which include upregulation of intestinal hydrolases and transporters during disaccharide (maltose) digestion [17], and sucrase inhibition by l-arabinose [18]. Currently, knowledge on digestion of rare sugars, such as kojibiose and nigerose, has primarily been obtained from experiments with rat intestinal extract, which suggest delayed digestion of these sugars compared to maltose [15]. Although delayed digestion of sugars may impact glycemic responses in vivo, this cannot be assumed for many rare sugars in the absence of data from models simulating intestinal digestion and absorption in humans. Following digestion, glucose is actively absorbed by the sodium-dependent sodium/glucose cotransporter member 1 (SGLT1), whereas the glucose transporter type 5 (GLUT5) facilitates passive uptake of fructose, after which both enter the blood stream via the glucose transporter type 2 (GLUT2) [19]. Thereafter, the liver acts as a central organ in sugar metabolism that takes up glucose and fructose via the hepatic GLUT2 transporter [20,21]. Unlike glucose, the metabolism of fructose largely takes place in the liver, apart from conversions in the small intestine [22,23]. Fructose can contribute to hepatic fat storage via both de novo lipogenesis and interference with hepatic β-oxidation, specifically [7,24]. However, many studies describing a negative impact of fructose on the liver were performed with supraphysiological fructose concentrations or in combination with excess energy [25]. Furthermore, it is not known whether high fructose concentrations could also cause NAFLD in humans [26]. Animal experiments indicate that a combination of fatty acids and fructose in particular causes oxidative stress and severe steatosis [27,28].

In this study, we aim to study how different rare and conventional sugars impact liver health with the hypothesis that liver impact is reduced with sugars that are inefficiently digested or metabolized. Effects of sugars on liver health were investigated as such, and in combination with dietary and circulating fatty acids. These treatments were applied to the hepatoma G2 (HepG2) cell line, which is frequently used to study the effect of nutrition on metabolic health parameters, such as liver fat accumulation and inflammation [29,30]. Intestinal sugar digestion and absorption was simulated in human intestinal colorectal adenocarcinoma cells (Caco-2 cells), which undergo spontaneous differentiation to enterocyte-like cells with relevant expression of various digestive enzymes and nutrient transporters [31]. To allow cellular interaction and to study effects of disaccharides on liver metabolism, we combined Caco-2 and HepG2 cells as a model that has successfully been used to study iron fluxes [32] and polyphenol absorption [33], although it was not previously used for sugar research. We optimized the coculture model to make it fit-for-purpose, i.e., to study the impact of rare and conventional sugars on intestinal permeability, mitochondrial energy production, metabolic health-related gene expression and hepatic fat accumulation upon sugar/fatty acid exposure. Thereby, this study was one of the first to provide information on the molecular impact of rare disaccharides, highlighting the importance of structure-specific effects.

## 2. Materials and Methods

### 2.1. Materials

HepG2 (https://www.atcc.org/products/hb-8065, accessed 28 January 2021) and Caco-2 (https://www.atcc.org/products/htb-37, accessed 28 January 2021) cells were obtained from ATCC (Manassas, USA). Cell culture reagents, including Dulbecco’s Modified Eagle Medium (DMEM), Phosphate Buffered Saline (PBS), non-essential amino acids, trypsin-EDTA and penicillin/streptomycin, were purchased from Gibco (FisherScientific; Merelbeke, Belgium). The XF base medium, the RNA 6000 Nano chip and High sensitivity DNA chip were obtained from Agilent (Machelen, Belgium). Fetal Bovine Serum (FBS) and Trypan Blue were obtained from VWR (Leuven, Belgium). Twenty-four-well plates and Transwell plates with 24-well inserts were obtained from Corning (Elcolab; Kruibeke, Belgium), whereas 96-well plates and cell culture flasks (25 and 75 cm^2^) were obtained from Greiner (Vilvoorde, Belgium). The rare glucobioses nigerose (α1-3, 86%) and kojibiose (α1-2, 99%) were synthesized according to in-house procedures [10,11], whereas trehalose (α1-1 glucobiose, 99% pure) was kindly provided by Cargill (Mechelen, Belgium). Glucose, fructose, maltose (α1-4 glucobiose, 99% pure), galactose, mannose, ribose, the polyol mannitol, resazurin, Lucifer Yellow, trishydroxymethylaminomethane sulforhodamine B (SRB), Trypan Blue and rat-tail collagen were all purchased from Sigma-Aldrich (Overijse, Belgium). Xylose, l-arabinose, l-glutamine and oleic acid were obtained from Merck (Darmstadt, Germany), whereas palmitic acid was purchased from Fluka (Buchs, Switserland). NaHCO_3_, glacial acetic acid and ethanol were obtained from Chem-Lab (Zedelgem, Belgium). AdipoRed and neutral red were purchased from Lonza (Verviers, Belgium) and Santa Cruz Biotechnology (Heidelberg, Germany) respectively. Trichloroacetic acid (TCA) was obtained from Acros Organics (Geel, Belgium) and RNeasy plus mini kits were purchased from Qiagen (Hilden, Germany). QuantSeq 3′ mRNA-Seq Library Prep Kits were purchased from Lexogen (Vienna, Austria). Finally, the Quant-it ribogreen RNA assay was purchased from Life Technologies (Grand Island, NE, USA). The chemical structure of all sugars in this study is visualized in Figure 1.

### 2.2. Cell Culture and Experimental Setups

Caco-2 and HepG2 cells were cultured in DMEM with 25 mM and 5.5 mM glucose, respectively, supplemented with 10% heat-inactivated and sterile-filtered FBS, 1% non-essential amino acids and 1% penicillin/streptomycin. Cells were incubated in a CO_2_ incubator (Memmert; VWR, Leuven, Belgium) at 37 °C and 10% CO_2_, and the cell medium was refreshed every two or three days. Cells were used between passages 5 and 30. At 80% confluency, cells were trypsinized and split at a ratio of 1:3 or 1:5, for Caco-2 and HepG2 cells, respectively. The cell suspension was mixed 1:1 with Trypan Blue and cells were counted using a Bürker counting chamber (VWR; Leuven, Belgium). Caco-2 and HepG2 cells were seeded at a density of 2 × 10^4^ cells per well in 96-well plates. Seeding and treatment conditions depended on the different setups visualized in Figure 2. 

#### 2.2.1. Monoculture Setup

Cells used for monocultures were seeded on clear 96-well plates at a density of 2 × 10^4^ cells per well. These cells were exposed for 24 h in a sugar-free XF base medium, to which sugars and fatty acids were added, as explained in Figure 2. Fatty acids were added as a solution in pure ethanol, and were diluted 400 times in the medium, resulting in a final non-toxic ethanol concentration of 0.5%. Fatty acid exposures were performed in the presence of 10% FBS, and a 1:1 mixture of palmitic and oleic acid was used for the final experiments in combination with the sugars.

Monoculture fat accumulation experiments were performed after 24-h exposure to 28 mM sugar in combination with the 0.5 mM palmitic/oleic fatty acid mixture (Figure 2a). These concentrations were based on test experiments with AdipoRed and neutral red assays covering a range of 0–112 mM for monosaccharides and 0–16 mM for fatty acids.

#### 2.2.2. Coculture Models

Caco-2 cells for coculture were seeded in a 24-well transwell insert and were differentiated for at least 3 weeks. Twenty-four-well plates were coated with 7.5 μg/cm^2^ collagen according to the manufacturer’s protocol [34], including an additional overnight UV treatment, and HepG2 cells for coculture were seeded at a density of 5 × 10^4^ cells per well. Differentiated Caco-2 cells and confluent HepG2 cells were cocultured for 24 h, followed by a 24-h exposure. Exposures for energy effects (28 mM sugar) and gene expression (55 mM sugar) were performed with 200 μL sugar solution, added to the apical side of the well, and 1 mL sugar-free medium added at the basolateral side (Figure 2b). We opted to perform a gene expression analysis with monosaccharides at a higher concentration of 55 mM, as this ensures sufficiently high sugar concentrations following intestinal transport and allows one to perform a sugar comparison at the HepG2 level, rather than the effect of different glucose concentrations due to delayed digestion of glucobiose exposures.

Fat accumulation in the coculture model was tested with a ‘free fatty acid’ (simulating circulating fatty acids) and ‘dietary fatty acid’ model (simulating fatty acids in the diet), which both combine sugars with 0.5 mM palmitic/oleic acid mixture. In the ‘free fatty acid model’, fatty acids were added to the basolateral compartment and sugars were added apically (Figure 2c). In the ‘dietary fatty acid model’, sugars and fatty acids were both added apically (Figure 2d). 

### 2.3. Intestinal Permeability

Intestinal permeability was measured by monitoring the transepithelial electrical epithelial resistance (TEER) [35] and by performing a Lucifer Yellow assay [36]. TEER was determined using an automated REMS device (World Precision Instruments; Sarasota, FL, USA) prior to the coculture initiation, after 24 h coculture and after exposure to sugars and fatty acids. TEER values were measured in Ω and divided by 3 to correct for the surface area of the 24-well transwell plates, presenting them as normalized Ω·cm^2^ values. Lucifer Yellow transport was measured by adding 0.1 mg/mL of Lucifer Yellow in the apical cell culture medium, and after 1 and 2 h of exposure, 200 μL of the basal medium was transferred to a black 96-well plate for fluorescence measurement (Spectramax M2 plate reader, Molecular Devices; Berkshire, United Kingdom), λ_exc_/λ_em_ = 428/540 nm. Lucifer Yellow transport was calculated based on a standard curve covering the range of 0–100 μg/mL. LOD and LOQ were 0.77 and 2.33 μg/mL, respectively.

### 2.4. Resazurin Assay for Cellular Reductase Activity

Resazurin stock solution (1 mg/mL in distilled water) was added to the cell medium at 1:100 *v/v* [37]. The plate was incubated for two hours at 37 °C and fluorescence was measured (λ_exc_/λ_em_ = 560/590 nm) in a black 96-well plate. 

### 2.5. AdipoRed Assay for Intracellular Fat Accumulation

Following 24 h of exposure with sugar and fat, HepG2 cells were washed with phosphate-buffered saline (PBS), and 2.5% of AdipoRed was added in the basolateral compartment. The plate was incubated for 15 min at 37 °C and bottom fluorescence was measured (λ_exc_/λ_em_ = 485/572 nm).

### 2.6. RNA Isolation

RNA from cocultured Caco-2 and HepG2 cells was isolated using the RNeasy plus mini kit from Qiagen. Cells were first washed with PBS and then lysed with 200 or 350 µL RLT lysis buffer for Caco-2 and HepG2 cells, respectively. The procedure from the manufacturer’s protocol was followed, and RNA was finally eluted in 40 μL RNase-free water.

### 2.7. RNA Processing and Sequencing

All samples had an RNA integrity number (RIN) value above 9. RNA from each sample was quantified using the ‘Quant-it ribogreen RNA assay’ (Life Technologies, Grand Island, NE, USA) and 500 ng RNA was used to prepare an Illumina sequencing library using the QuantSeq 3′ mRNA-Seq Library Prep Kit (Lexogen, Vienna, Austria) according to manufacturer’s protocol with 14 enrichment polymerase chain reaction (PCR) cycles. An average of 9.0 × 10^6^ ± 1.8 × 10^6^ and 11.6 × 10^6^ ± 1.0 × 10^6^ reads was generated for HepG2 and Caco-2 samples, respectively. 

### 2.8. Neutral Red Assay

Neutral Red working solution was prepared by diluting neutral red stock solution 1:80 in serum-free DMEM. Cells were washed with PBS (Ca and Mg), followed by a 3 h incubation with 150 μL/well of neutral red working solution. The cells were washed again, and 100 μL of desorb solution (50% ethanol, 49% distilled water and 1% glacial acetic acid) was added. After 20 min, absorbance was measured at 540 nm, as explained in Repetto et al. [38].

### 2.9. Sulforhodamine B Assay for Protein Content

Energy production and intracellular fat accumulation were corrected for protein content by performing a Sulforhodamine B (SRB) assay [39]. After the assays, cells were fixated with 1:4 *v/v* 50% TCA in medium for at least one hour at 4 °C. The cells were washed with tap water at least 3 times and SRB solution was added in excess. After 30 min, the plate was washed at least 3 times with 1% glacial acetic acid. Next, the protein-adhered SRB stain was absorbed by adding 200 μL 10 mM Tris and pipetted up and down to homogenize the stain. Absorbance was measured at 490 nm.

### 2.10. Statistics and RNA Data Analysis

Statistics were used to determine differences between mean values for the total pool of well replicates of an exposure condition, generated from multiple different plates that were seeded from the same cryovial at different passages. Results from ‘optimization experiments‘ based on a single plate, were qualitatively described without the use of statistics. Statistical analyses were performed with SPSS 26 using a significance cut-off of *p* < 0.05. Levene’s tests were performed to check for homogeneity of variance. Conditions were compared with one-way analysis of variance (ANOVA), using the Tukey’s correction for homogeneous data or a Welch ANOVA with a Games–Howell correction for non-homogenous data.

Analysis for differential gene expression was performed using the edgeR’s [40] quasi-likelihood method between 2 conditions, only including genes that were expressed at a counts-per-million (cpm) above 1 in at least 5 samples. Genes were considered significantly differential if they had a false discovery rate (FDR) < 0.05, as well as a fold change of at least 2. Gene set enrichment analysis (GSEA) was performed using the GAGE R package [41], based on Kyoto encyclopedia of genes and genomes (KEGG) pathways provided by this package. Genes and KEGG pathways of interest were selected based on their impact on diabetes pathology, hepatic fat synthesis and the energy metabolism (Appendix A). 

## 3. Results

We first present the model development and defined the sugar and fatty acid conditions to obtain physiologically relevant effects. To this end, we tested viability to define the sub-toxic range and determined which physiological exposures induce cellular responses. Next, we tested the individual as well as the combined/synergistic impact of sugars and fatty acids at multiple concentrations on fat accumulation in liver cells.

### 3.1. Effect of Sugars and Fatty Acids on Lysosomal Activity

To define which concentrations of sugars and fats are tolerated by the different cell types, neutral red assays for lysosomal activity were performed.

No decrease in neutral red incorporation in HepG2 cells was observed upon sugar exposures within the tested concentration range (0–112 mM, Appendix A). Exposure to the mixture of palmitic and oleic acid for 24 h was accompanied by lower neutral red incorporation at fatty acid concentrations of at least 1 mM in HepG2 cells or 8 mM in Caco-2 cells (Appendix A). We then used subtoxic concentrations of maximal 0.5 mM fatty acids and 55 mM sugars in other assays.

### 3.2. Hepatic Fat Accumulation

#### 3.2.1. Effects of Sugars in the HepG2 Monoculture Model

First, we aimed to simulate the lipogenic effect of fructose, as reported in vivo [42]. To this end, fructose, as such, and in the presence of palmitic and oleic acid, which are described to stimulate fat accumulation in liver cells [43], were added to the HepG2 cells.

Visually higher protein-corrected AdipoRed fluorescence as indication for intracellular fat accumulation was observed in an optimization experiment at increasing concentrations of a 1:1 palmitic/oleic acid mixture. Exposure to 28 mM fructose as such did not induce fat accumulation in the optimization experiment, but appeared to enhance the effect of the fatty mixture at 0.5 mM (Figure 3a).

In a sugar comparison experiment, the effect of 0.5 mM fatty acid mixture was increased significantly by 28 mM glucose, fructose and galactose, but not by mannose, xylose, ribose and l-arabinose (Figure 3b). 

#### 3.2.2. Effects of Sugars in the Caco-2/HepG2 Coculture Models

In a second step, we extended this model with intestinal brush border digestion and absorption. After 24-h apical exposure (dietary fatty acid model) to a 0.5 mM palmitic/oleic acid mixture and 28 mM monosaccharide or 14 mM disaccharide, fat accumulation responses were highly variable within conditions and not significantly different between conditions. 

The effect of a fatty acid mixture as such was limited, whereas exposure to a combination of fatty acids and monosaccharides (especially fructose) resulted in a large but non-significant (*p* = 0.09) median increase in fat accumulation (Figure 3c). In comparative experiments with disaccharides, a smaller but equally variable and non-significant effect (*p* = 0.08) was observed for maltose, nigerose and kojibiose, but not for trehalose (Figure 3d). 

In the free fatty acid setup with direct contact between liver cells and fatty acids, a strong stimulating effect of the fatty acid mixture on fat accumulation was observed, which appeared to be enhanced by glucose in a proof-of-concept experiment (Figure 3c), but was not significantly impacted by glucobioses during a comparative analysis (Figure 3d).

### 3.3. Cellular Physiology of Cocultured Caco-2 and HepG2 Cells

The 24-h coculture of HepG2 and Caco-2 cells in glucose-containing medium did not influence TEER (Appendix A). In contrast, 24-h exposure to mannitol (−172 Ω·cm^2^, 516 Ω or −34%) or l-arabinose (−245 Ω·cm^2^, 735 Ω or −43%) at 55 mM in sugar-free medium resulted in a significant decrease of TEER (Figure 4a), although TEER values remained above 325 Ω·cm^2^ (975 Ω), and this decrease of TEER was not accompanied by increased Lucifer Yellow transport (vales remained below LOQ). TEER values were maintained in cells exposed to glucose, fructose, kojibiose or nigerose (Figure 4a). 

Resazurin conversion was significantly increased by 24-h exposure to 14 mM nigerose in Caco-2 cells, and by maltose, kojibiose (both 14 mM) and glucose (28 mM) in HepG2 cells (Figure 4b). HepG2 cells exposed to these four sugars had a 35% to 55% higher protein-corrected resazurin conversion compared to the control, whereas trehalose did not have an effect.

### 3.4. Gene Expression Analysis

To investigate whether absorbed monosaccharides were able to modify hepatic responses, gene expression was measured in Caco-2 and HepG2 cells from the coculture model, and exposed to glucose, fructose, arabinose and mannitol. After 24-h exposure to 55 mM of glucose, fructose, l-arabinose and mannitol, differential gene expression was visible between all exposures in HepG2 cells, but not in Caco-2 cells (Table 1). In HepG2 cells, glucose altered gene expression most, followed by l-arabinose (Table 1 and Appendix A). Appendix A show the principle component analysis (PCA) plots for all conditions compared to glucose and between the most differential conditions (glucose versus l-arabinose). Several of the genes that were downregulated by glucose compared to the mannitol control were significantly upregulated by l-arabinose instead. On the pathway level, significant differences were observed between all exposures in both HepG2 and Caco-2 cells (Table 2 and Figure 5). 

To facilitate the interpretation of the gene expression results in the metabolic context, the impact of the tested sugars on key genes and KEGG pathways involved in digestion, energy metabolism and metabolic health is visualized in Figure 5 and Figure 6. Figure 5 summarizes the pathways, whereas Figure 6 illustrates how these effects impact metabolism and metabolic health. Figure 6a shows that the oxidative phosphorylation pathway, as an important pathway for ATP production, was significantly more expressed in glucose versus l-arabinose exposed Caco-2 and HepG2 cells. In the HepG2 cells, both fructose and l-arabinose upregulated the hypoxia-inducible factor 1 (HIF1) pathway that inhibits the oxidative metabolism. 

Figure 6b shows that glucose, fructose and l-arabinose all provide distinct effects on metabolic health pathways in HepG2 cells. Glucose strongly upregulated the hepatic thioredoxin-interacting protein (TXNIP), arrestin domain containing 4 (ARRDC4), the pathway for peroxisome proliferator-activated receptor (PPAR) signaling and downstream pathways, such as the pathways for biosynthesis of fatty acids and steroid hormones (Figure 5b and Appendix A). Glucose downregulated the tumour necrosis factor (TNF), rat sarcoma (RAS) and mitogen-activated protein kinase (MAPK) pathways and numerous pro-inflammatory genes, such as interleukin-11 (Il-11), early growth response protein 1 (EGR1) and Fos proto-oncogene (c-FOS). Compared to l-arabinose exposed cells, glucose-exposed cells also had a higher expression of acetyl-CoA carboxylase (ACC), as the rate-limiting enzyme for fatty acid synthesis, and a lower expression of sirtuin-1 (SIRT1), as the longevity gene that inhibits the pathways of PPARα and ү.

Fructose upregulated TXNIP less strongly than glucose, and uniquely upregulated the metabolically important insulin signaling and AMP-activated protein kinase (AMPK) pathways. The pathway for type 2 diabetes was significantly more active in Caco-2 cells exposed to fructose, specifically, compared to those exposed to glucose.

l-Arabinose upregulated the hepatic expression of the phosphatidylinositol 3-kinase/Protein kinase B serine/threonine protein kinase (PI3K/AKT) and MAPK pathways downstream of insulin signaling, whereas the advanced glycation end products receptor for the advanced glycation end products (AGE-RAGE) pathway involved in diabetes complications was upregulated by l-arabinose in both HepG2 and Caco-2 cells. 

## 4. Discussion

In this paper, we described effects of rare/alternative (trehalose, nigerose, kojibiose and l-arabinose) versus conventional sugars (glucose, fructose, galactose and maltose) on energy metabolism, gene expression, intestinal permeability and liver fat accumulation in Caco-2/HepG2 (coculture) setups. The major achievements are (I) the use of the Caco-2/HepG2 model for the novel application of studying sugar metabolism, (II) a distinction between (rare) disaccharides based on their impact on energy production (III), the demonstration of the impact of monosaccharide structure on liver metabolism, liver fat accumulation and gene expression.

### 4.1. Trehalose, Not Kojiniose or Nigerose, Has Less Metabolic Impact Than Conventional Glucobioses

An important finding was that glucobioses differentially impact energy provision in HepG2 cells, following their digestion in Caco-2 cells. Whereas maltose and kojibiose provided similar energy effects as glucose in the HepG2 cells, trehalose did not provide any stimulatory effects, suggesting that trehalose is cleaved more slowly into glucose. Experiments with rat intestinal extract also reported that trehalose is the most slowly digested glucobiose [15]. The study of Lee et al. has also demonstrated that small intestinal enzymes maltase, glycoamylase, isomaltase and sucrase have the highest affinity for maltose with the α1-4 bond, a lower affinity for nigerose (α1-3) and kojibiose (α1-2), and no affinity for trehalose (α1-1), which is a substrate for the enzyme, trehalase [15]. As a result of its slower digestion rate, trehalose had a smaller glycemic response compared to conventional sugars in a human intervention study [58]. Our findings confirm the reduced digestibility and metabolic impact of trehalose compared to conventional sugars, and even compared to other rare glucobioses. However, the reduced cellular impact of trehalose may be related to the expression of brush border enzymes in Caco-2 cells. Sucrase-isomaltase was the dominantly expressed brush border enzyme in our Caco-2 cells (according to the gene expression data) with a much higher expression than trehalase and lactase-phlorizin hydrolase, which is in line with a study reporting the activities of these disaccharidases in Caco-2 cells [59]. Nevertheless, trehalase activity in the human intestine is also considerably lower compared to sucrase and especially maltase activity [60], indicating that low trehalase expression is not necessarily a limitation of our model.

Disaccharide digestion and related cellular energy production may have impacted TEER values in Caco-2 cells, as a selective TEER drop in cells exposed to mannitol or l-arabinose was observed, suggesting that energy provision is required to maintain TEER. l-Arabinose is described as a zero calorie sugar that contributes to weight loss in mice with the metabolic syndrome [61], which could be linked to our gene expression data, where the oxidative phosphorylation pathway in l-arabinose exposed HepG2 and Caco-2 cells was significantly less active compared to glucose-exposed cells. Previously, it has been reported that ATP depletion induced by a derivative of ‘carbonyl cyanide phenylhydrazone’ (CCCP) results in a dose-dependent increase of Caco-2 monolayer permeability [62]. The TEER drop in low-energy conditions may not be problematic for the coculture model, as the Lucifer Yellow and gene expression data indicated no damage to the tight junctions, and Caco-2 cells cultured with fetal bovine serum have a higher electrical resistance than our small intestine in vivo [35]. However, these findings may be a second indication for efficient glucose release from kojibiose and nigerose. 

Nevertheless, glucose release from the glucobioses may be insufficient to enhance hepatic fat accumulation, as the significant effect of glucose in the monoculture setup was not observed for easily digestible glucobioses in the coculture setups. This may also be related to the large variation within conditions, which could be explained by variability introduced by cross-talk in coculture models as well as by exposures in sugar-free medium that cause some cells to switch to more aerobic substrates, such as fatty acids, while others remain on a highly glycolytic metabolism (depending on the added sugar). Additionally, substantial variation is observed specifically in the ‘dietary fatty acid setup’ due to smaller fat-induced responses in the absence of a direct fatty acid exposures, and therefore larger percentual fluctuations in fat accumulation responses. In the less variable ‘free fatty acid setup’, the combination of an indirect exposure to a digestible sugar and a direct exposure to fatty acids, provides a fat-induced response that may be too dominant to be enhanced by glucose released during digestion. The trend towards higher median fat accumulation with energy providing disaccharides without showing significant effects, may suggest that longer exposure durations are required for more substantial effects to occur, especially because all glucobioses but trehalose provided increased fat accumulation responses in individual wells. This could also be an indication that the effects of sugars are more subtle in more realistic models (such as the Caco-2/HepG2 coculture setups) without direct exposure of hepatic cells to high sugar concentrations. 

Overall, we conclude (I) that trehalose behaves as the least digestible and therefore most promising (in the context of sugar replacement, to investigate further in search for low-glycemic sugars) glucobiose in Caco-2 cells, and (II) that glucobioses with significant stimulatory effects on cellular energy production provide less consistent effects on hepatic fat accumulation.

### 4.2. Monosaccharides Differentially Impact Hepatic Fat Accumulation and Gene Expression

In contrast to disaccharides, monosaccharide-specific effects were observed on hepatic fat accumulation, where only glucose, fructose and galactose increased fat accumulation. Although fructose is often described as a uniquely lipogenic sugar, the enhancing effects of both glucose and fructose, are in line with other studies testing fat accumulation in monocultures with HepG2 cells challenged with glucose, fructose, fatty acids and insulin [29,63]. NAFLD is a disease with an important genetic component with a heritability of about 30% in population-based studies and higher percentages in twin-studies, depending on whether twins grew up in the same environment [64]. In our study, there were no sugar-induced changes in gene expression for the most common genes linked to NAFLD in genome-wide association studies (GWAS). However, there were sugar-induced epigenetic changes that may impact hepatic fat accumulation. The upregulation of fatty acid biosynthesis by glucose may further support a lipogenic effect of glucose, especially as the rate-limiting enzyme for fat synthesis, acetyl-CoA carboxylase (ACC) [65], was significantly higher-expressed in glucose-exposed compared to l-arabinose-exposed cells. These differential effects of glucose and l-arabinose on hepatic ACC expression can be linked to another study, in which an l-arabinose intervention in rats with the metabolic syndrome reduced hepatic ACC mRNA expression [61]. In contrast, we did not observe these direct effects on fatty acid biosynthesis pathways for fructose, which may be partially explained by the absence of fatty acids in the gene expression experiments. As fructose may interfere with beta-oxidation in the presence of fatty acids and hence increases hepatic fat accumulation [66], fructose may not need to stimulate fatty acid biosynthesis genes in order to provide adverse effects, especially in the presence of fatty acids. Effects of fructose on hepatic fat accumulation may also be mediated by uric acid, which is produced during the fructose metabolism, induces fat accumulation in the liver and may interfere with mitochondrial function [67,68,69]. In addition, increased substrate availability due to a fast fructose conversion to acetyl-CoA/GAP may have contributed to liver fat accumulation, and was suggested by Hirahatake et al. to be a more important contributor to fructose-induced lipogenesis than transcriptional alteration of lipogenic enzymes [70]. It should be noted that the high sugar exposures performed directly to the HepG2 may be less representative for the human situation compared to the coculture setups with an indirect exposure via the intestinal Caco-2 cells. Therefore, the larger median fat accumulation with fructose in our dietary fatty acid setup may be an indication that this sugar, in particulae, could have an effect in vivo, despite the smaller impact on gene expression. 

In addition to fatty acid metabolism and lipogenesis, we observed monosaccharide-specific effects on gene expression related to glucose homeostasis and diabetes pathology in HepG2 cells especially, where the glucose altered pathways that could eventually contribute to diabetes-related problems. These epigenetic findings are important considering the involvement of both genetic and environmental factors in type 2 diabetes that contribute to a lifelong risk of 40% and 70% for the development of type 2 diabetes for people with one or two diabetic parents, respectively [71]. Although expression of TCF7L2 as gene that is most consistently linked to type 2 diabetes in GWAS was not affected in our study [72], the glucose modulated the expression of important genes and pathways for type 2 diabetes. Firstly, the strong upregulation of hepatic TXNIP and ARRDC4 by glucose, may contribute to insulin resistance, considering that these two genes are insulin pathway inhibitors that can be activated by the glucose/fructose responsive transcription factor MondoA and thereby contribute to the adverse effects of MondoA on fat accumulation and insulin resistance [73]. In addition, TXNIP is involved in the regulation of β-cell function, cellular glucose uptake and hepatic glucose output, overall providing adverse effects on glucose homeostasis and acting as a potential therapeutic target for diabetes [54,74]. Moreover, the upregulation of fatty acid biosynthesis and other PPAR-related pathways, suggest an more active lipid and cholesterol metabolism in glucose-exposed HepG2 cells overall. In contrast, downregulation of the TNF pathway and the il-11, FOS and EGR1 genes by glucose may be beneficial for reducing inflammation and preventing vascular dysfunction, respectively [75]. Although EGR1 expression is usually upregulated by high glucose exposure in endothelial cells, this has not been observed in hepatocytes in other studies either [75,76]. Alternatively, the lower expression of inflammatory genes in the presence of glucose could be interpreted as a pro-inflammatory effect of glucose depletion, which is a phenomenon that is observed with hypoglycemia in vivo [77].

Fructose may have impacted metabolic health via the AGE-RAGE pathway in HepG2 cells, as a pathway related with diabetes pathology and vascular complications [78,79]. This effect on the AGE-RAGE pathway may be plausible, as fructose is known to stimulate advanced glycation end product (AGE) formation even more than glucose [80]. In contrast, upregulation of the insulin signaling and AMPK pathways by fructose may be a more beneficial epigenetic alteration, considering the central role that AMPK plays in maintaining metabolic health via its impact on insulin secretion, gluconeogenesis, lipogenesis and its anti-inflammatory action [81,82]. 

Surprisingly, the non-metabolizable l-arabinose had an impact on gene expression as well. Unlike the situations where glucose provided an effect and l-arabinose did not, effects on the 33 genes that were altered by l-arabinose compared to the mannitol control cannot be explained by glucose retraction, as the mannitol control did not contain glucose either. Previously, Osaki et al. already reported that l-arabinose as a sucrase inhibitor prevents lipogenic gene expression effects of sucrose [53]. Furthermore, Zhao et al. reported that l-arabinose provides protective effects on both the physiological and epigenetic level in rats on a high-fat diet [83]. Their findings suggested that l-arabinose reduces the negative impact of the high-fat diet by improving lipid oxidation and thermogenesis via modulation of gene- and protein expression. Furthermore, these researchers reported that l-arabinose enhances the stimulatory effect of the high-fat diet on fat catabolism and the inhibitory effect on fat anabolism [83]. However, it is new to observe epigenetic effects of l-arabinose in the absence of metabolic challenges, such as high sucrose or fat conditions. The upregulation of both the PI3K/AKT and MAPK pathway downstream of insulin signaling in our study potentially suggests that l-arabinose improved insulin sensitivity. SIRT1, as gene that was expressed more in cells exposed to l-arabinose compared to glucose, inhibits lipid synthesis, while also acting as insulin sensitizer [84]. On the other hand, l-arabinose upregulated the AGE-RAGE pathway in both Caco-2 and HepG2 cells, which should be interpreted with caution, as there is no literature available suggesting that l-arabinose would adversely impact AGE production. 

Overall, we can conclude that monosaccharides provide differential effects on both the physiological and epigenetic level. l-Arabinose provided more beneficial effects compared to glucose and fructose on the physiological level, whereas the health benefits were less clear on the epigenetic level. 

### 4.3. Strengths, Limitations and Future Perspectives

This study performed an improved comparison between rare and conventional glucobioses, which has previously only been performed using rat intestinal extract [85], which solely provides information on glucose release. The cellular approach allows one to measure the actual impact of the different flow of monosaccharides released from the disaccharides. Moreover, our coculture model allows one to study the metabolic effects of rare and conventional disaccharides and links intestinal digestion and absorption with the post-absorptive hepatic metabolism, thereby providing clear benefits over monoculture models in terms of relevance for simulating the complex in vivo situation. In addition, we successfully combined sugars and fatty acids. The fatty acid mixture in our study is frequently used in cellular research, and incorporates an inflammatory component (palmitic acid) and a fatty acid that is more easily stored in liver cells (oleic acid) [43]. Finally, the transcriptomics approach in the coculture setup allowed further investigation of the mode-of-action, i.e., regarding lipogenesis and insulin sensitivity.

However, the experimental approach has limitations as well, starting with the single-timepoint approach. Gene expression of receptors and transports involved in sugar digestion are strongly time-dependent, which means that a single timepoint cannot capture all effects [17]. We have chosen the 24-h timepoint as a timepoint that is relevant for many metabolic responses and matches the time required to observe effects in the other assays. As gene expression analysis does not provide information on altered protein production, confirmation with proteomics would strengthen the results [86]. Furthermore, it should be mentioned that sugar concentrations of 28 mM used in the HepG2 monoculture exceed normal blood glucose levels [87], and primarily serve as proof of concept that metabolic abnormalities can be induced by specific sugars in this cell model. All concentrations (0–55 mM) used in the coculture setup are intestinal exposures within the physiological range [88]. The decision to use sugar-free medium for the HepG2 cells in the coculture (instead of a background of 3–5 mM glucose, as normally present in the blood stream) was based on the potent effects of even small amounts of glucose on cellular processes, such as energy production, which hinders the evaluation of digestible sugars in cell culture models. Next, there are limitations related to the cell models, such as the absence of the maltase-glucoamylase complex in Caco-2 cells, as confirmed by our gene expression data [17]. However, maltase-glucoamylase is responsible for only 20% of the maltase activity in humans compared to 80% for the sucrase-isomaltase complex [16], and Caco-2 cells are described to have high maltase activity despite the absence of maltase-glucoamylase [59]. Lastly, we did not measure or quantify digestion and transport of sugars from the Caco-2 to the HepG2 cells in this exact model, while knowledge on rates of digestion and transport would strengthen the findings. 

In the future, trehalose may be the most interesting glucobiose to serve as a sugar replacer, as it was the only glucobiose without any stimulatory effect on energy production or fat accumulation in the liver, with the note that trehalase expression was low in Caco-2 cells. As l-arabinose provided no adverse effects on the physiological level, alters some gene expression pathways and is described to limit the negative effects of sucrose, l-arabinose may both be promising as a sugar replacer and as an addition to the diet. However, our conclusions are based on a few aspects of the larger metabolic story, which means that more metabolic health aspects of individual sugars should be investigated. Furthermore, additional molecular research focusing on underlying mechanisms, post-translational alterations and knock-out models could contribute to a better understanding of our results. Therefore, metabolomics and research on how rare sugars influence development of insulin resistance may help to identify the healthiest sugars. 

## 5. Conclusions

Little is known about the cellular effects of rare- and non-conventional sugars upon digestion and absorption, especially regarding their impact on metabolic health and related health disorders, such as fatty liver disease and type 2 diabetes. In a Caco-2/HepG2 model to investigate the post-absorptive effects of sugars, we observed differential effects of glucobioses on energy production and monosaccharide-specific effects on hepatic fat accumulation and gene expression, highlighting the potential of specific non-conventional sugars, such as trehalose and l-arabinose, whereas nigerose and kojibiose had more impact on energy production. These findings demonstrate that sugar exposures at the intestinal level can cause differential post-absorptive effects and further support the importance of sugar-specific effects beyond the distinction between rare- and conventional sugars. Based on these findings, we suggest that (rare) sugars should be treated by health experts as structure-specific nutrients with an individual health impact rather than as one group with predominantly adverse health effects. If also true in vivo, this may strongly affect the current health communication towards the consumer, and the food labeling of ‘sugars’ in the future. Future research could include the aspect of insulin resistance, confirm epigenetic changes on the protein level and explore mechanisms further with knock-out models.

## Figures and Tables

**Figure 1 nutrients-14-00611-f001:**
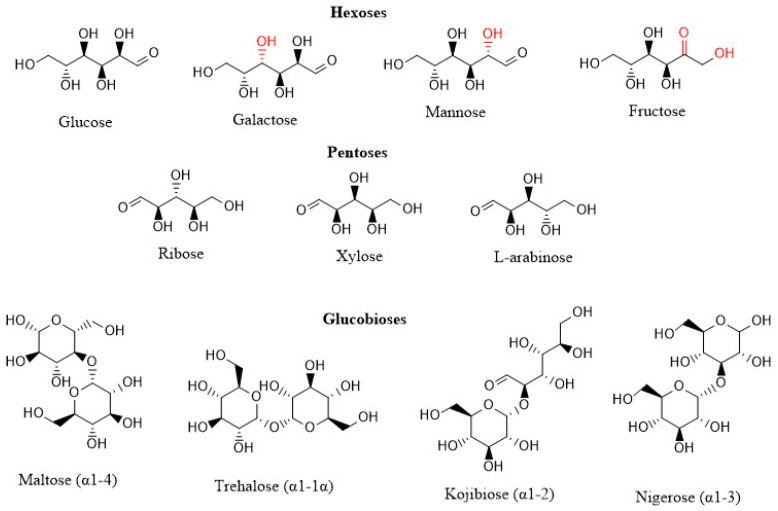
Chemical structure of sugars that were investigated in this study (ChemDraw). Differences in chemical structure compared to glucose are indicated in red (for hexoses). Glucobioses differ in chemical bond indicated behind the name of the molecule.

**Figure 2 nutrients-14-00611-f002:**
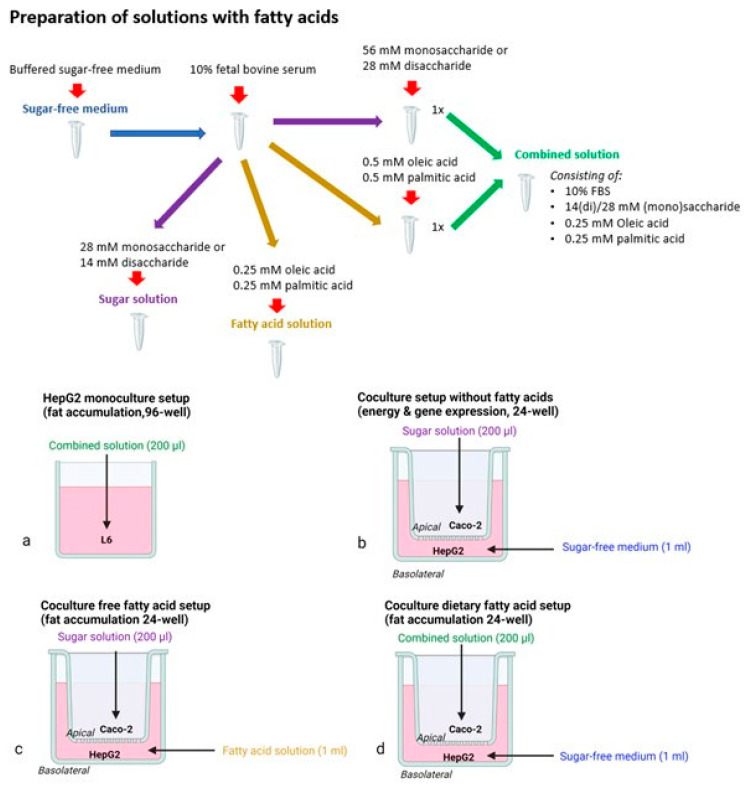
Experimental setups and exposure preparations (Biorender). These setups include a monoculture setup (**a**) and coculture setup: without fatty acids (**b**), free fatty acid setup (**c**) and a dietary fatty acid setup (**d**).

**Figure 3 nutrients-14-00611-f003:**
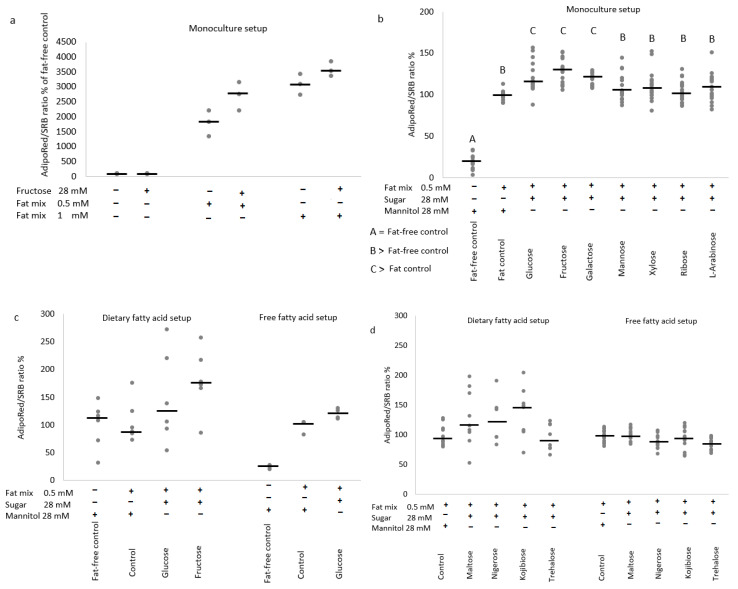
The effect of 24-h sugar exposures on protein-corrected AdipoRed fluorescence in monoculture (**a**,**b**) and coculture setups (**c**,**d**) with HepG2 cells. Protein-corrected AdipoRed fluorescence as measure for intracellular fat accumulation is shown as % relative to the fat-free or fat-containing control. (**a**,**c**) Are proof-of-concept figures demonstrating the effects of a 1:1 palmitic/oleic mixture and enhancing effect of glucose or fructose in the different setups (generated from a single plate experiment with 3 wells per condition for the free fatty acid setup or from 2 independent plates for a total of 6 wells per condition for the dietary fatty acid setup). (**b**,**d)** Demonstrate the effects of different monosaccharides at 28 mM in combination with 0.5 mM palmitic/oleic acid mixture in the monoculture setup (**b**), 18-well replicates spread over 3 experiments with independent plates, or different disaccharides at 14 mM + fatty acids in the dietary fatty acid setup (**d**), 10-well replicates spread over 3 experiments with independent plates and free fatty acid setup (**d**), and 15-well replicates spread over 4 experiments with independent plates. Data are presented as individual responses + median value stripe (**c**,**d**), with different capital letters indicating statistically significant differences between conditions (*p* < 0.05).

**Figure 4 nutrients-14-00611-f004:**
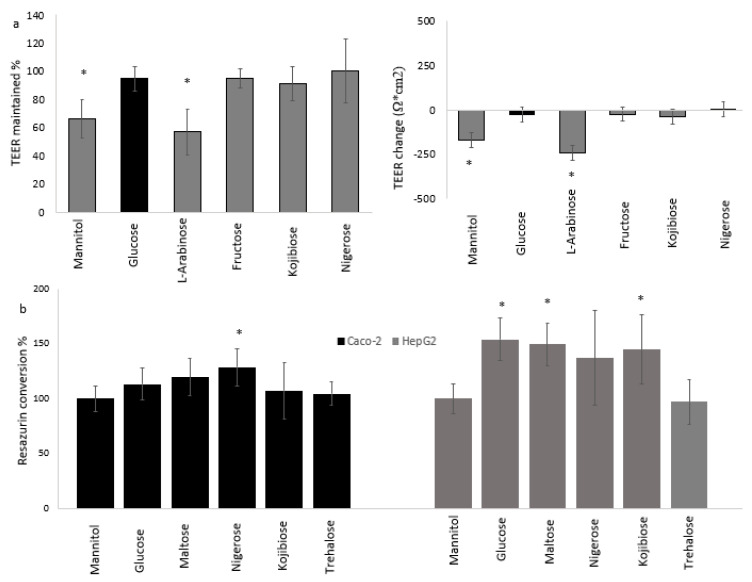
The effect of 24-h sugar exposures in sugar-free medium on TEER and resazurin conversion. (**a**) shows the effect of 55 mM monosaccharides or half the concentration for disaccharides on TEER in (as % maintained or absolute change compared to glucose) in Caco-2 cells. (**b**) shows the effect of 28 mM monosaccharides or half the concentration for disaccharides on resazurin conversion (shown as % relative to the mannitol control). Data were generated from experiments with 3 independent plates, generating a total of 11 (**a**) or 10 (**b**) well-replicates per condition, and are presented as mean ± standard deviation with * indicating a statistically significant (*p* < 0.05) difference compared to glucose.

**Figure 5 nutrients-14-00611-f005:**
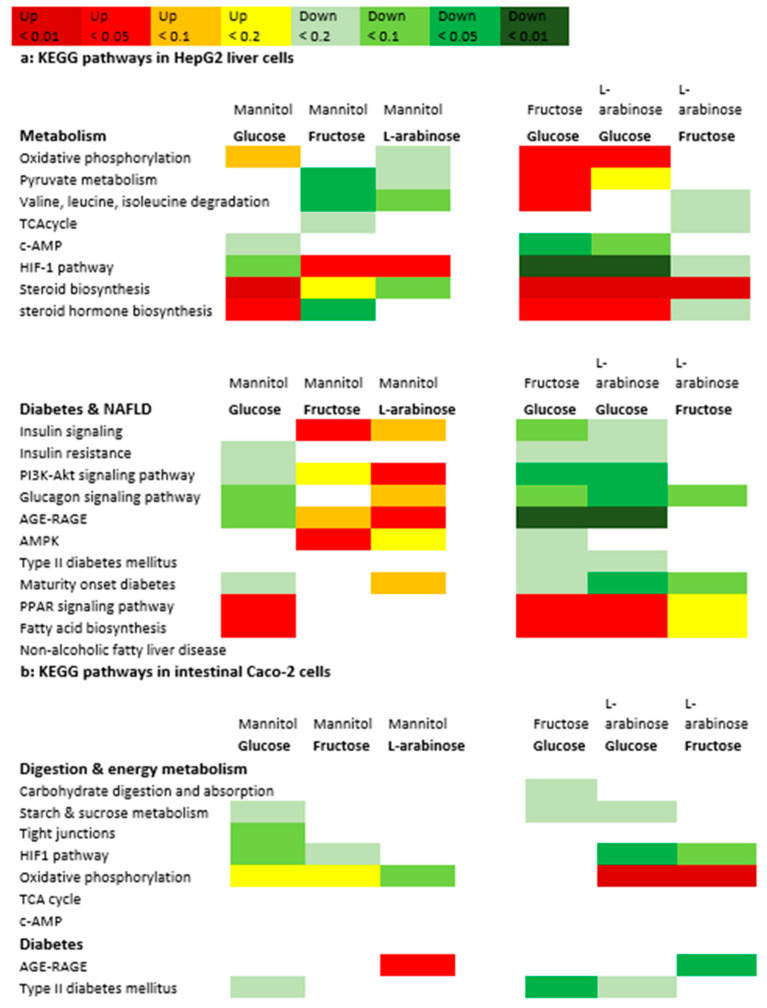
Overview of effects of glucose, fructose and l-arabinose on expression of relevant KEGG pathways in HepG2 (**a**) and Caco-2 cells (**b**).

**Figure 6 nutrients-14-00611-f006:**
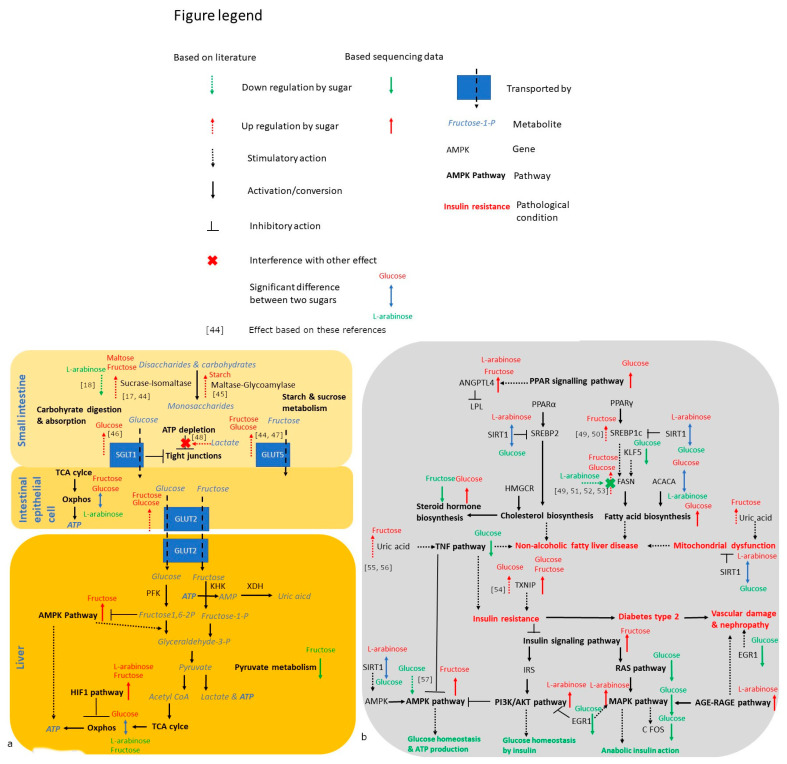
Pathway figure showing how energy metabolism (**a**) and metabolic health (**b**) are affected by 55 mM glucose, fructose and l-arabinose (5 well-replicates spread over 3–5 independent plates per condition). Bold arrows next to a gene or pathway indicate that they are significantly (*p* < 0.05 following corrections for multiple comparison) impacted by 24-h sugar exposure: green (downregulation compared to mannitol), red (upregulation compared to mannitol) or blue (significantly higher for the sugar stated above versus below the arrow) [17,18,44,45,46,47,48,49,50,51,52,53,54,55,56,57].

**Table 1 nutrients-14-00611-t001:** Number of genes and KEGG pathways that are altered by the different sugars. Differential genes between exposures.

Differential Genes per Cell Line	Glucose Mannitol	Fructose Mannitol	l-Arabinose Mannitol	Glucose Fructose	Glucose l-Arabinose	Fructose l-Arabinose
HepG2	89	10	33	6	376	10
Caco-2	0	0	1	0	2	0

**Table 2 nutrients-14-00611-t002:** Number of genes and KEGG pathways that are altered by the different sugars. Differential pathways between exposures.

Fructose Mannitol	l-Arabinose Mannitol	Glucose Fructose	Glucose l-Arabinose	Fructose l-Arabinose
41	53	113	100	33
8	10	2	17	16

## Data Availability

Gene expression data is available in the Gene Expression Omnibus: https://www.ncbi.nlm.nih.gov/geo/query/acc.cgi?acc=GSE192753 (accessed on 28 January 2021). Other data is included in the article submission to Nutrients. Samples generated from our cell lines are registered in a database: ‘Biobank Vakgroep Levensmiddelentechnology, Voedselveiligheid en Gezondheid’ with accesscode BB190156.

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
