# Peer review of "Metabolism and Health Effects of Rare Sugars in a CACO-2/HepG2 Coculture Model"

_nutrients, 2022, doi:10.3390/nu14030611_

Round 1
Reviewer 1 Report
please see attachment

Reviewer 2 Report
Non-alcoholic fatty liver disease (NAFLD) is a widespread disease. In their study, Amar van Laar et al. describe the effects of rare (trehalose, nigerose, kojibiose, L-arabinose) sugars compared to conventional carbohydrates (glucose, fructose, galactose, maltose) on energy production, intestinal permeability, liver fat accumulation and gene expression. They observed different structure-specific effects, suggesting health potential for trehalose and L-arabinose.
This work has some merit because it is one of the first report, which provides relevant information on the molecular impact of rare disaccharides. The manuscript is well written.
Some minor points should be considered:
- Page 3, first paragraph: “24 well plates and…” should be rewritten to “Twenty-four well plates and…” or “Overall 24 well plates and…”
- Page 4, last paragraph: “24-Well plates…” should be rewritten to “Twenty-four well plates…” or “Overall 24 well plates and…”
- Page 6, Subtitle: “2.9. SRB assay….” Should be spelled out to “Sulforhodamine B assay…”
Author Response
Dear reviewer,
We appreciate your input and have implemented your suggestions.
They can be found at lines 109, 158 and 218.
All other changes can be seen in the 'track changes file'.
Round 2
Reviewer 1 Report
The rebuttal is convincing and I regard the revised manuscript suitable for publication. One last comment: p-values should be "0.05" instead of "0,05".